# Continuous respiratory rate monitoring through temporal fusion of ECG and PPG signals

**Yuxuan Lin**[1], **Xinyue Song**[1], **Yan Zhao**[1], **Chunlin Zhang**[1], **Xiaorong Ding**[1,2]*

1 School of Life Science and Technology, University of Electronic Science and Technology of China, Chengdu, Sichuan, China, 2 Yangtze Delta Region Institute (Huzhou), University of Electronic Science and Technology of China, Huzhou, Zhejiang, China

* xiaorong.ding@uestc.edu.cn

**Data availability statement:** The third party data used for this study are publicly available

## Abstract

Respiratory rate (RR) is an important vital sign indicating  various pathological conditions, such as clinical deterioration, pneumonia, and adverse cardiac arrest. Traditional RR measurement methods are normally intrusive and inconvenient for ubiquitous continuous  monitoring. There have been studies on RR estimation by extracting respiratory modulated components (RMCs) from wearable accessible noninvasive cardiovascular signals, such as electrocardiogram (ECG) or/and photoplethysmogram (PPG), with RR estimated from each RMC or fused RMCs derived from either ECG or PPG. However, there is  few study  on robust continuous RR estimation with the combination of all kinds of RMCs from both ECG and PPG in the time domain. In this study, we propose the temporal fusion of RMCs extracted from both ECG and PPG signals to estimate RR with the aim to improve  estimation performance. We extracted six RMCs from ECG and PPG, identified those RMCs of high quality with the respiratory quality index, fused the identified ones into one respiratory signal with principal component analysis, and estimated the RR from the fused signal. Validation on two public datasets - the Capnobase dataset (42 subjects) and the BIDMC dataset (53 subjects) - showed that the proposed method attained a mean absolute error (MAE) of 1.39 breaths/min and 3.29 breaths/min for RR estimation, respectively, achieving an average 11.61% reduction in MAE compared to existing state-of-the-art approaches. This demonstrates that temporal fusion of the RMCs of wearable ECG and PPG can improve the performance of RR estimation.

## 1 Introduction

Respiratory rate (RR) is a key vital sign that can indicate various pathological conditions like acute clinical deterioration, adverse cardiac events, and respiratory disorders, including respiratory dysfunction, pneumonia, and acute respiratory distress syndrome [1,2]. RR is also a key index used for assessing intensive care needs and predicting mortality [3]. Continuous, noninvasive, and real-time monitoring of RR is important for the detection of early deterioration and evaluation of the risk of acute respiratory disorders.

from Capnobase
(https://doi.org/10.5683/SP2/NLB8IT) and
BIDMC
(http://doi.org/10.1109/TBME.2016.2613124).

**Funding:** This work was supported by Huzhou
ST Special Program of Huzhou (2023GZ01) and
the National Natural Science Foundation of
China (82102178). The funders had no role in
study design, data collection and analysis,
decision to publish, or preparation of the
manuscript.

**Competing interests:** The authors have
declared that no competing interests exist.

Traditional respiratory monitoring methods, such as spirometry, capnography, and lung compliance assessment, are intrusive, cumbersome, and discomforting, making them unsuitable for continuous monitoring. Over the past two decades, various new approaches have been explored for indirect, continuous RR monitoring using unobtrusive sensors [4]. The studies [5,6] discuss physiological monitoring systems and wearable sensors, which provide a basis for the development of unobtrusive and non-invasive RR monitoring. Wearable electrocardiogram (ECG) and photoplethysmogram (PPG) signals are frequently studied due to their interaction with the respiratory and circulatory systems. Those studies include direct RR detection from filtered ECG or PPG signals [7–9]and time-frequency analysis-based RR monitoring methods [10–12]. Recent advancements in signal processing algorithms have significantly enhanced computational efficiency [13–15]. Therefore , neural network-based methods for RR estimation have emerged [16–19], though they often fail to explain the underlying physiological mechanisms of respiration.

Many studies have aimed to improve the accuracy and robustness of RR monitoring through signal fusion. The challenges in multi-signal fusion methods lie in selecting the appropriate signals and developing effective fusion strategies. As shown in Fig 1, the fusion strategies of the signal fusion methods are categorized into three types based on the fusion stage: *i) Early Fusion*: direct fusion at the wearable signals level (e.g., ECGs, PPGs, etc.); *ii) Intermediate Fusion*: fusion after extracting respiratory components or respiratory features; and *iii) Late Fusion*: combining multiple RR estimates using mean, median, or weighted average.

To our knowledge, few studies have focused on fusion at the wearable signal level, with more attention given to late fusion strategies for RR monitoring. Researchers have typically estimated RR from respiratory modulated components (RMCs) or features extracted separately from ECG or PPG, then fused the multiple RR estimates using mean, median, or weighted averages [20–22]. The limitation of late fusion methods is that combining RR estimates at the final stage leads to significant data loss and fails to fully utilize the respiratory-related information in ECG or PPG signals.

The respiratory and circulatory systems interact through mechanical, hemodynamic, and neurohumoral pathways [23]. Fusion from RMCs can effectively preserve respiration-related information in ECG and PPG signals. However, Khreis *et al.* [24] showed that RMCs are highly dependent on individual conditions and activity status. The direct fusion of all RMCs, considering these complexities, may not enhance RR monitoring accuracy. Thus, a targeted fusion strategy is required to combine RMCs tailored to different individuals. Applying the

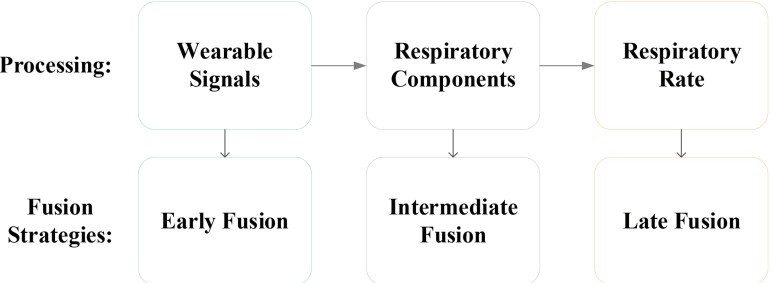

**Fig 1. Fusion strategies.** Three types of signal fusion strategies are categorized based on the different stages of signal fusion: Early Fusion, Intermediate Fusion, and Late Fusion.

Respiratory Quality Index (RQI) to assess the quality of extracted RMCs before fusion can help address this issue [25].

Respiratory component fusion strategies can be classified into frequency-domain and time-domain fusion. In frequency-domain fusion, Lázaro *et al.* [26] computed the mean power spectrum of multiple respiratory component signals to derive respiratory frequency. Pimentel *et al.* [27] estimated RR by fusing the auto-regressive spectra of three respiration-induced changes: frequency, amplitude, and intensity. Chan *et al.* [28] applied modality-attentive fusion to denoise ECG and PPG signals for RR estimation during walking. However, frequency-domain methods are complex and often assume signal stationarity, limiting their performance with non-smooth signals like ECG and PPG. Given the significant RR variation in ambulatory monitoring, time-domain fusion may better capture transient RR changes [29].

This study proposes a method based on the temporal fusion of RMCs extracted from both ECG and PPG. To reduce noise impact, we filtered, evaluated, and screened the RMCs. A time-domain fusion approach was then employed to better capture RR variations. Ultimately, RR monitoring was performed using the fused respiratory waveforms, aiming to improve accuracy in dynamic monitoring. The main contributions of this study are:

(1) We employed both PPG and ECG signals to extract respiration-related components and subsequently fuse them in the time domain. This approach has the potential to yield more comprehensive respiratory-related information compared to the fusion of multiple respiratory rate estimations;

(2) We evaluated the proposed method on two publicly available datasets, Capnobase and BIDMC, with experimental results demonstrating its superior performance over both single-modality approaches and frequency-domain fusion methods.

## 2 Datasets and methods

As illustrated in Fig 2, the proposed RQI-PCA method systematically integrates five core phases: (1) multi-source RMCs extraction, (2) RMCs filtering, (3) optimal subspace screening through RQI screening mechanism, (4) temporally adaptive fusion of selected components, and (5) robust RR estimation. Algorithm 1 further formalizes this workflow.

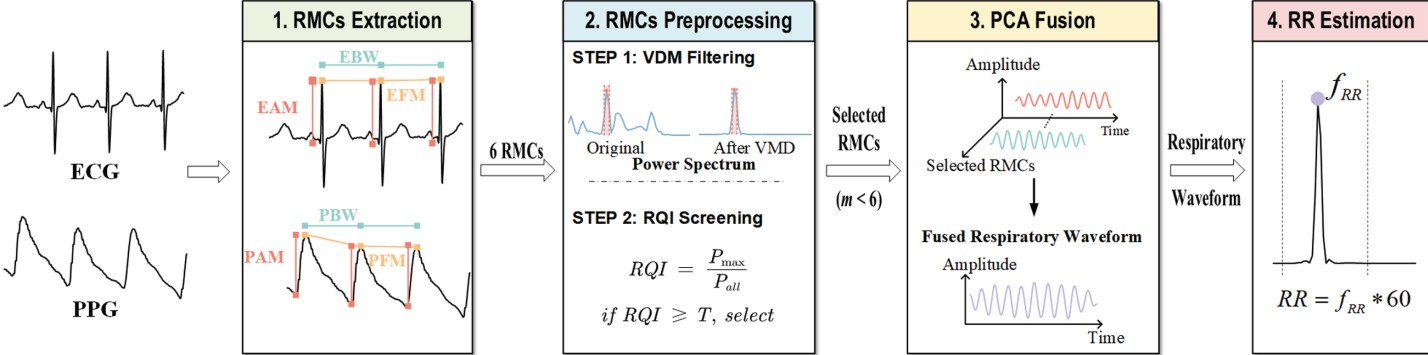

**Fig 2. Outline of the proposed method.** Outline of the proposed approaches: 1. respiratory modulated components (RMCs) extraction: three modes of RMCs, i.e., amplitude modulation - AM, frequency modulation - FM, and baseline wander - BW, were extracted from ECG and PPG, which gives the ECG / PPG AM (EAM, PAM), ECG / PPG FM (EFM, PFM) and ECG / PPG BW (EBW, PBW); 2. RMCs Preprocessing: denoising of the RMCs with variational mode decomposition (VMD) and screening of the RMCs with respiratory quality index (RQI), where T represents the average RQI of all 6 RMCs; 3. principal component analysis (PCA) fusion of the screened RMCs; 4. Respiratory rate(RR) estimation from the fused respiratory waveform.

**Algorithm 1.** Respiratory Rate Estimation via Temporal Fusion of ECG and PPG Signals

> **Input:** Raw PPG and ECG signals, Processing window length $W_R = 30s$
> **Output:** Estimated respiratory rate $RR_{est}$ (breaths/min)

1 Resample, filter, and segment ECG & PPG signals into $N_R$ non-overlapping windows of length $W_R$;
2 **for** $n = 1 \rightarrow N_R$ **do**
3 Calculate six respiratory-modulated components $\{RMC_i\}_{i=1}^{6}$ from ECG and PPG signals;
4 **for** $i = 1 \rightarrow 6$ **do**
5 Compute power spectral density (PSD) of $RMC_i$ denoted as $P_{RMC_i}(f)$;
6 Identify dominant frequency $f_M(i) = \arg\max_{f \in [f_{min}, f_{max}]} P_{RMC_i}(f)$;
7 Decompose $RMC_i$ into five intrinsic mode functions (IMFs) using variational mode decomposition(VMD): $RMC_i = \sum_{j=1}^{5} IMF_j$;
8 **for** $j \rightarrow 5$ **do**
9 Compute $IMF_j$'s PSD $P_{IMF_j}(f)$ and peak frequency $f_j$;
10 **if** $f_j = f_M(i)$ **then**
11 select $IMF_j$ as refined $RMC_i$;
12 **end**
13 **end**
14 Calculate the respiratory quality index (RQI) of $RMC_i$ (denoted as $RQI_i$) via Equation (1);
15 **end**
16 Compute mean RQI across all windows: $T = \frac{1}{6} \sum_{i=1}^{6} RQI_i$;
17 **for** $i = 1 \rightarrow 6$ **do**
18 **if** $RQI_i \geq T$ **then**
19 select $RMC_i$;
20 **end**
21 **end**
22 Perform principal component analysis (PCA) on selected $RMC_i$ to fuse components into a continuous waveform $RW_{est}$;
23 Compute the PSD of $RW_{est}$;
24 Identify dominant frequency $f_{RR} = \arg\max_{f \in [f_{min}, f_{max}]} P_{RW_{est}}(f)$;
25 Convert to breaths/min: $RR_{est} = f_{RR} \times 60$.
26 **end**

## 2.1 Dataset

To verify the generalization performance of the algorithm, we used two databases, Capnobase and BIDMC, for our experiments to cover a wider range of clinical scenarios. The Capnobase dataset contains ECG, PPG, and respiratory signals taken simultaneously during elective surgery and routine anesthesia in 42 subjects over a period of 8 minutes (all sampled at 300 Hz) [30]. There were 23 cases of spontaneous breathing and 19 cases of controlled.

The BIDMC database [27] includes data from 53 critically ill patients during hospitalization. Each subject's data contains age, gender, and physiological signals - PPG, impedance respiratory signal and ECG - all sampled at 125 Hz. Physiological parameters such as heart rate, RR (manually annotated by two specialists), and oxygen saturation were also recorded as reference values. Data was recorded for 8 minutes per case.

## 2.2 Extraction of respiratory modulated components

The raw signal was segmented into 30-second windows to balance accuracy and computational efficiency. Firstly, the PPG and ECG signals for each 30-second window were denoised. After resampling at 300 Hz, a 10-point smoothing filter was applied to attenuate high-frequency noise. A 0.05 Hz third-order Butterworth high-pass filter was then applied to remove low-frequency noise. Subsequently, three typical respiratory components - baseline wander, amplitude modulation, and frequency modulation - were extracted from both ECG and PPG signals.

For ECG signals, the Pan-Tompkins algorithm [31] was used to identify the R-peak, with the Q wave defined as the minimum value occurring with a 200ms interval preceding the R wave. As shown in Fig 2 three distinct RMCs were delineated [21]:*i) ECG Amplitude Modulation (EAM)*: the amplitude difference between Q and R wave within a cardiac cycle, caused by cardiac rotation and thoracic impedance; *ii) ECG Frequency Modulation (EFM)*: the temporal interval between successive R-peaks, influenced by respiratory sinus arrhythmia; *iii) ECG Baseline Wander (EBW)*: the average amplitude of the Q and R wave within a cycle, with a physiological mechanism similar to EAM.

For PPG signals, peak detection was influenced by reflected waves, so a threshold method was utilized to ascertain the PPG troughs. The maximum value within a 125 ms window after the trough was designated as the peak of the systolic period. Based on this, three RMCs were proposed [21]: *i) PPG Amplitude Modulation (PAM)*: the amplitude difference between the onset and end of the systolic period within a PPG cycle, driven by cardiac output modulation; *ii) PPG Frequency Modulation (PFM)*: the time interval between the systolic peaks of adjacent PPG cycles, influenced by respiratory sinus arrhythmia; *iii) PPG Baseline Wander (PBW)*: the midpoint amplitude at the start and end of the systolic period within a cycle, affected by intra-thoracic pressure variations.

## 2.3 Respiratory modulated components preprocessing

**2.3.1 Respiratory components filtering using VMD.** RMCs contained noise and other frequency information, so variational mode decomposition (VMD) was applied to further purify the signals. VMD is an adaptive signal processing technique that decomposes a signal into intrinsic mode functions (IMFs) [32]. Each IMF represents a distinct frequency scale with relative stability. This method is particularly effective for processing respiratory signals, offering greater robustness than empirical mode decomposition (EMD) and ensemble empirical mode decomposition (EEMD) [33].

RMCs have a narrow frequency band and fewer harmonics, which belong to the "spectrum sparse" signal, and a small number of IMFs can be selected. Through systematic parametric optimization experiments, we determined that the VMD level of 5 optimally balances the extraction of the signal's primary time-frequency components while avoiding mode mixing (see Fig 3). This configuration ensures the preservation of the original signal's physical interpretability. Subsequently, we calculated the maximum peak frequency within the power spectrum of each IMF. The IMF with the smallest absolute deviation from the peak frequency of the input signal's power spectrum was selected as the representative filtered respiratory component. Fig 3 shows an example of VMD filtering, the spectral peak of IMF3 (0.3 Hz) exhibits exact correspondence with the characteristic frequency of the original signal. In contrast, IMF1 and IMF2 predominantly consist of high-frequency noise components, whereas IMF4 and IMF5 represent low-frequency residual artifacts. Based on this spectral alignment and component characterization, IMF3 was rigorously identified as the critical modality for subsequent analysis, ensuring fidelity to the signal's intrinsic features.

**2.3.2 Respiratory signal screening using RQI.** The RMCs are dependent on the patient's condition and activity status, making it challenging to predict in advance which signal will be effective for particular individual components. Therefore, before estimating RR for different individuals, signal quality evaluation of each RMC is required. This study employed the RQI for signal quality evaluation.

RQI evaluates the quality of RMCs by assessing the regularity of respiratory peaks and the periodicity of respiratory waveforms. It quantifies respiratory-related information within the modulated components extracted from PPG or ECG [25]. The standard RQI calculation

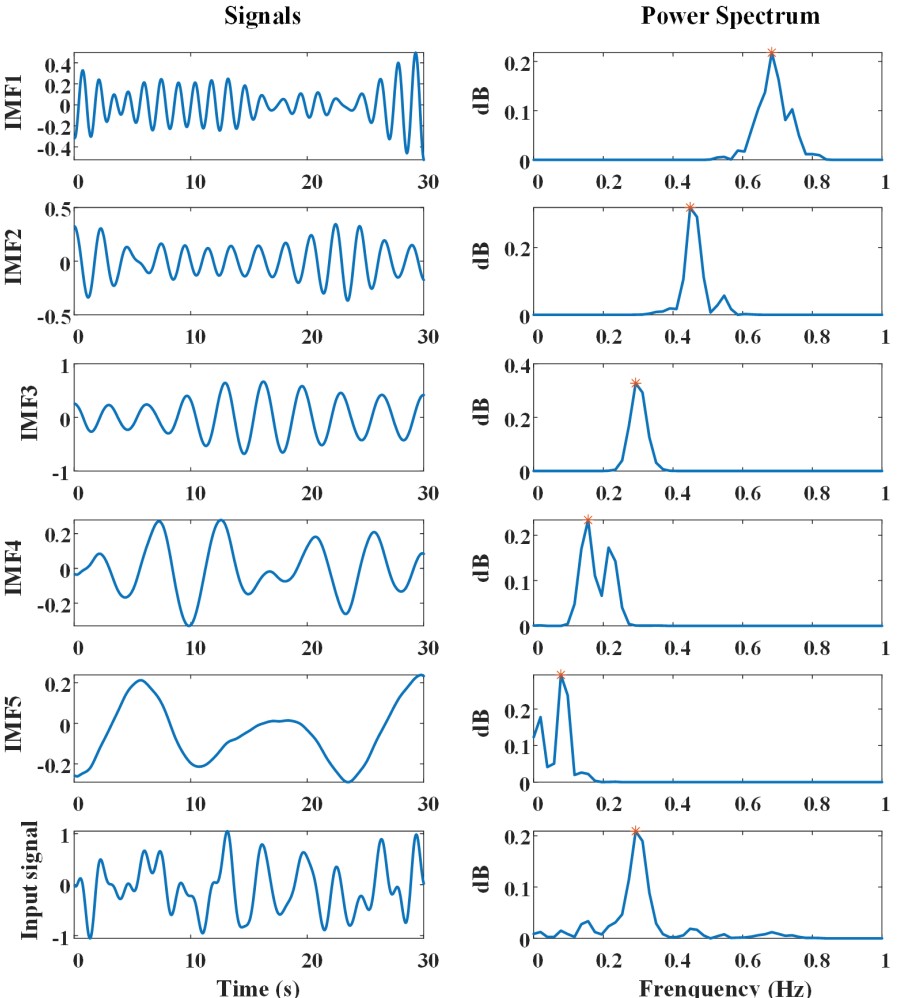

**Fig 3. Signal denoising by variational mode decomposition (VMD).** Left panel: input original signal and the decomposed intrinsic mode functions (IMFs); Right panel: the corrresponding power spectrum of the signals.

uses Fourier transform to define the energy concentration of a specific frequency band of the RMC [34]:

$$RQI = \frac{P_{max}}{P_{all}},\tag{1}$$

where $P_{max}$ is the sum of the spectral coefficients within the physiological respiration range of 0.05 to 0.75 Hz (equivalent to 3 - 45 breaths/min), and $P_{all}$ is the sum of Fourier transform coefficients across the entire spectrum. The ratio of $P_{max}$ to $P_{all}$ evaluates the proportion of respiratory component information, as shown in Fig 2. A higher RQI value indicates more energy concentration in the RMC within the respiratory frequency range. For each analysis window, the average RQI of all six modulated components, denoted as $T$, served as the adaptive threshold. $m$ respiratory components with RQI $\geq T$ were selected for subsequent fusion.

## 2.4 Temporal respiratory signal fusion

A regular breathing pattern can be approximated by a sinusoidal waveform, where the dominant frequency corresponds to the RR frequency [35]. In respiratory signal processing, reducing the dimensionality of multiple respiratory signals is important to minimize redundancy while retaining the dominant frequency characteristics.

Previous studies have confirmed that methods such as principal component analysis (PCA), t-distributed stochastic neighbor embedding [36], the Gaussian process latent variable model [37], and diffusion map [38] can effectively fuse signals. Among these, PCA proved to be the most computationally efficient, making it ideal for integration into compact wearable devices. Consequently, PCA was chosen to fuse the multi-time respiratory series.

PCA can reduce the M-dimensional information in X space to K-dimensional information in Y space (K<M) with less information loss. Assuming that there are $m$ RMCs after RQI screening, the steps of PCA fusion are as follows:

(1) Form matrix **X** by arranging the $m$ RMCs as columns.

(2) Center the matrix **X** (mean subtraction).

(3) Compute the covariance matrix **P** and calculate its eigenvalues and eigenvectors.

(4) Sort the eigenvectors into a matrix in descending order of their corresponding eigenvalues, then take the first row to form matrix **Q**.

(5) Compute **Y** = **QX**, which yields the fused respiratory waveform $RW_{est}$.

Then, the fused respiratory waveform was analyzed using its power spectrum. Within the respiratory frequency range [0.05, 0.75] Hz, the maximum peak frequency was denoted by $f_{RR}$, and the RR was calculated as $RR = f_{RR} * 60$ breaths/min.

## 2.5 Performance measurement

To assess the accuracy and consistency of the estimated RR, we conducted a comprehensive analysis using several statistical and error metrics. Correlation coefficient scatter plots were used to assess the linear relationship with reference values. Bland-Altman plots assessed the agreement between the two signal screening methods. Estimation errors were then quantified using the following metrics:

- Mean Error (ME):

$$ME = \frac{\sum_{i=1}^{n} \left( RR_{est}(i) - RR_{ref}(i) \right)}{n}. \tag{2}$$

- Standard Deviation (STD):

$$STD = \sqrt{\sum_{i=1}^{n} \frac{\left[ \left( RR_{est}(i) - RR_{ref}(i) \right) - ME \right]^2}{n-1}}. \tag{3}$$

- Mean Absolute Error (MAE):

$$MAE = \frac{\sum_{i=1}^{n} \left| RR_{est}(i) - RR_{ref}(i) \right|}{n}. \tag{4}$$

- Root Mean Square Error (RMSE):

$$RMSE = \sqrt{\frac{\sum_{i=1}^{n} \left( RR_{est}(i) - RR_{ref}(i) \right)^2}{n}}. \tag{5}$$

In these equations, $RR_{est}$ represents the estimated RR, and $RR_{ref}$ signifies the reference RR.

## 3 Results

### 3.1 Preprocessing and temporal fusion

Firstly, six RMCs were identified by locating the inflection points (peaks and troughs) in the ECG and PPG signals. Fig 4 shows the EAM, EFM, EBW, PAM, PFM, and PBW extracted from a typical subject's single window in the Capnobase dataset. The RMCs extracted from the PPG signal, particularly PAM and PBW, exhibited more pronounced periodicity.

Then, the six RMCs were filtered using VMD. Fig 5(a)–(d) display the original and denoised RMCs along with power spectra. As illustrated in Fig 5(b) and (d), the target frequency band of the filtered signal power spectrum was clear, and VMD effectively reduced the noise and irrelevant information.

Finally, the clean RMCs selected by RQI were fused to obtain an estimated respiratory signal (as shown in Fig 5(e)). The respiratory waveform estimated by RQI-PCA closely resembles the reference respiratory waveform. Compared with the single respiratory modulation signal, the signal after fusion reduced the frequency of false wave peaks and made the waveform smoother.

### 3.2 Performance of RR estimation

RRs were estimated using the proposed method for 42 samples from the Capnobase database. The results of the significance and consistency analyses are presented in Fig 6. Most data points fell within the 95% confidence interval and in close proximity to the zero line. The scatter plots and Bland-Altman plots showed a strong correlation and consistency between the estimated and reference RR.

We computed the estimation errors for each individual RMC, all ECG RMCs (EAM, EFM, EBW), all PPG RMCs (PAM, PFM, PBW) and all ECG & PPG RMCs using the RQI-PCA

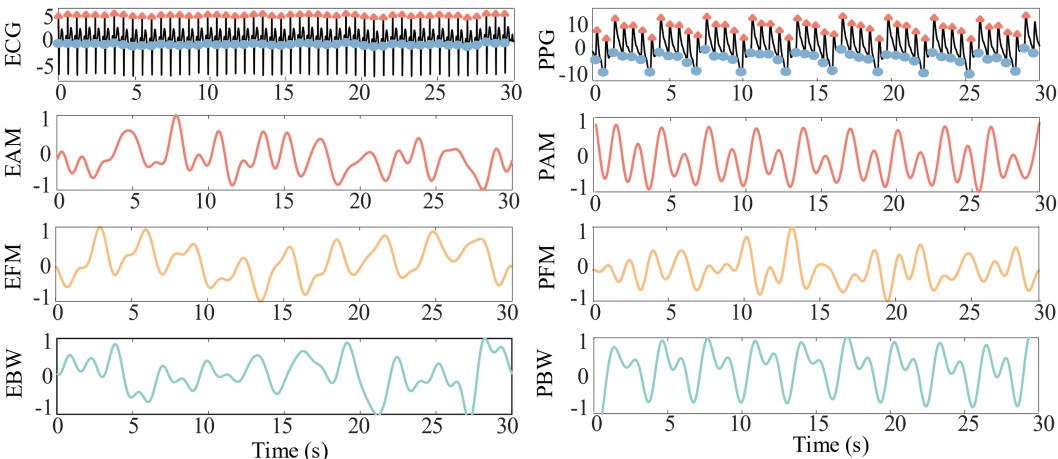

**Fig 4. Respiratory modulated components extraction.** Representative electrocardiogram (ECG) and photoplethysmogram (PPG) signals alongside the corresponding respiratory modulated components extracted. ECG Amplitude Modulation (EAM), ECG Frequency Modulation (EFM), ECG Baseline Wander (EBW), PPG Amplitude Modulation (PAM), PPG Frequency Modulation (PFM), and PPG Baseline Wander (PBW).

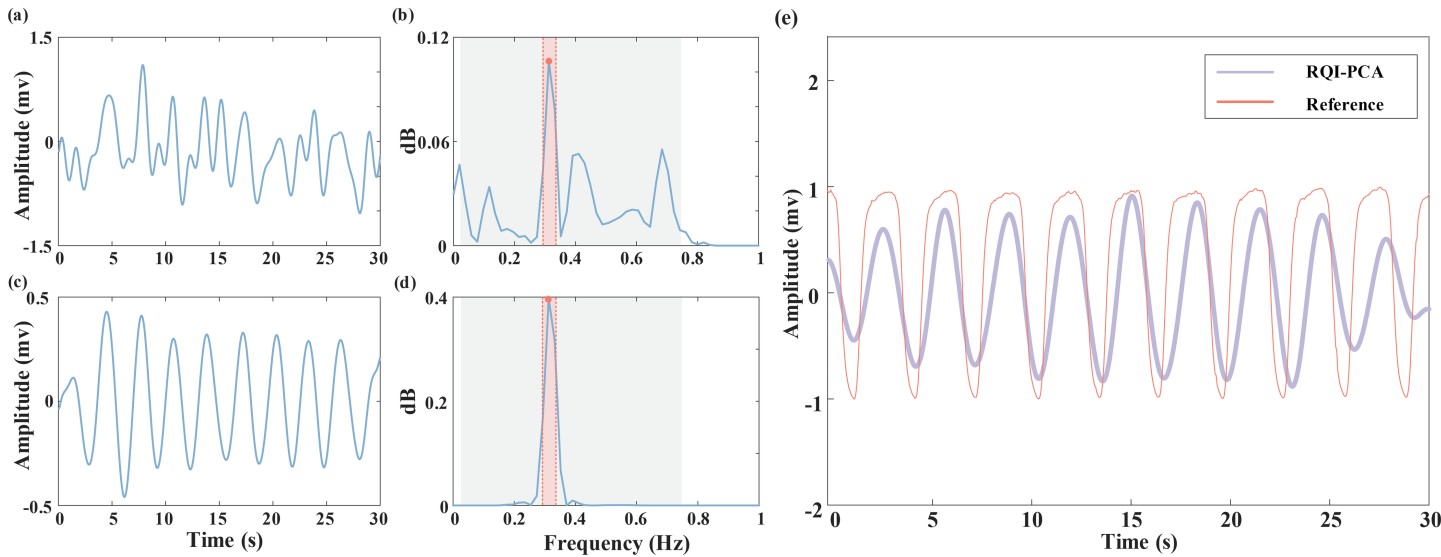

**Fig 5. Respiratory modulated components filtering and fusion.** Comparison of respiratory modulated components (RMCs) before and after variational mode decomposition (VMD), as well as the comparison of estimated respiratory signal and reference respiratory signal: (a) Time domain waveform of the RMC before VMD; (b) Frequency domain power spectrum of the RMC signal before VMD; (c) Time domain waveform of the RMC after VMD; (d) Frequency domain power spectrum of the RMC after VMD; (e) Estimated respiratory waveform obtained by fusing multiple RMCs and the reference respiratory waveform.

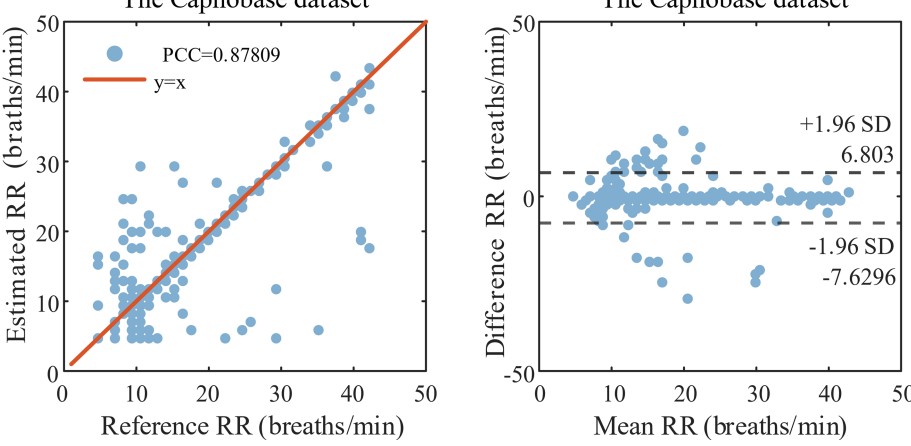

**Fig 6. Comparison of estimated respiratory rates and reference values.** Scatter plot (a) and Bland-Altman plot (b) of respiratory rate (RR) estimation with the proposed method (RQI-PCA) versus reference RR in Capnobase dataset.

method. Table 1 presents the estimation error from the Capnobase dataset. Among individual RMCs, EAM had the highest accuracy, followed by EFM and PBW. Consistent with Charlton's study [39], ECG signals generally outperformed PPG signals in RR estimation. Despite marginally lower ME values using the EFM and EBW individually, incorporating multiple RMCs improved other metrics. Although the fusion of all three ECG RMCs resulted in a lower ME (- 0.17 breaths/min), it performed worse on other metrics. In contrast, fusing six RMCs yielded consistent, robust performance. Therefore, temporal fusion methods based on six respiratory components provide more accurate predictions than methods using ECG or PPG alone. This finding aligns with C. Ahlstrom's earlier work [40].

**Table 1. Respiratory rate estimation performance in Capnobase.**

| No. | Components | ME ± STD (breaths/min) | RMSE (breaths/min) | MAE (breaths/min) |
|---|---|---|---|---|
| 1 | EAM | 0.74 ± 5.16 | 5.21 | 1.89 |
| 2 | EFM | 0.27 ± 6.56 | 6.56 | 3.29 |
| 3 | EBW | -0.29 ± 7.46 | 7.46 | 4.13 |
| 4 | PAM | 2.73 ± 7.74 | 8.2 | 4.96 |
| 5 | PFM | 1.45 ± 8.03 | 8.15 | 4.28 |
| 6 | PBW | 0.97 ± 6.59 | 6.66 | 3.34 |
| - | **No.1-3** | **-0.17 ± 4.17** | **4.17** | **1.60** |
| - | **No.4-6** | **0.48 ± 6.17** | **6.18** | **2.97** |
| - | **No.1-6** | **-0.41 ± 3.69** | **3.70** | **1.39** |

Performance comparison for respiratory rate estimation in Capnobase dataset using different respiratory modulated components with the RQI-PCA method.

The Capnobase database includes both controlled and spontaneous breathing states. We applied the RQI-PCA method, along with other comparison approaches: EEMD with spectral data fusion (Chung *et al.*) [41], EEMD followed by PCA (Motin *et al.*) [42], and PCA-based extraction of respiratory signals from the QRS trend in ECG (Langley *et al.*) [43] to estimate RR. Among these, only the Langley *et al.* method did not involve fusion. The Chung *et al.* method and Motin *et al.* method estimated RR by fusing IMFs from the PPG signal.

Fig 7 shows RR estimation time series for two subjects (0147 and 0332) from the Capnobase database. Subject 0147 was a 16-year-old adolescent with controlled intraoperative respiratory, while subject 0332 was a 39-year-old patient in a spontaneous breathing state. In controlled breathing, RR remains steady, with all methods, except Langley *et al.* and Motin *et al.*, providing highly accurate estimations (Fig 7(a)). However, in spontaneous breathing, our method exhibited superior predictive accuracy compared to other methods. Fig 7(b)

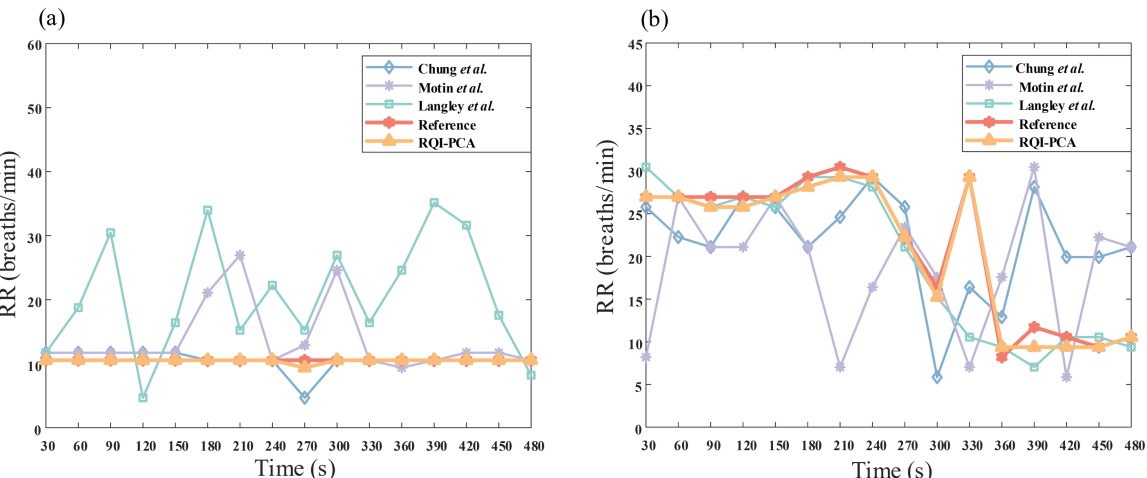

**Fig 7. Time-series of respiratory rate obtained using different methods.** Time-series of respiratory rate (RR) estimates for two representative subjects in Capnobase dataset with the methods including Chung *et al.* (blue line with diamond dots), Motin *et al.* (purple line with asterisk dots), Langley *et al.* (green line with square dots) and the proposed method (RQI-PCA) (yellow line with triangle dots) versus the reference values (red line with hexagonal dots): (a) subject 0147 in controlled breathing state and (b) subject 0332 in spontaneous breathing state.

**Table 2. Comparison between the proposed method (RQI-PCA) and methods in the previous literature with different signals.**

| Method | Datasets | Signals | Estimation Strategy | MAE (breaths/min) | RMSE (breaths/min) |
|---|---|---|---|---|---|
| Iqbal *et al.* [8] | BIDMC | PPG | Non-Fusion | 3.97 | - |
| Reddy *et al.* [45] | BIDMC | PPG | Non-Fusion | 3.95 | - |
| Lazazzera *et al.* [44] | Capnobase | PPG | Non-Fusion | 2.24 | - |
| Pimentel *et al.* [27] | BIDMC | PPG | Intermediate Fusion (Autoregressive Models, RMCs) | 4 | - |
| Motin *et al.* [46] | Capnobase | PPG | Intermediate Fusion (Principal Component Analysis, IMFs) | 3.3 | - |
| Karlen *et al.* [20] | Own | PPG | Late Fusion (SmartFusion, RRs) | - | 3.00 |
| Birrenkott et al. [34] | Capnobase | PPG & ECG | Intermediate Fusion (Linear Regression, RQIs) | 1.98 | - |
| Zhao *et al.* [49] | Capnobase & BIDMC | PPG & ECG | Intermediate Fusion (Deep Learning, respiratory futures) | 1.2 | - |
| Kumar *et al.* [50] | Capnobase | PPG & ECG | Intermediate Fusion (Deep Learning, respiratory futures) | 0.54 | - |
| Ding *et al.* [48] | Capnobase | PPG & ECG | Late Fusion (SmartFusion, RRs) | - | 1.76 |
| **RQI-PCA** | **Capnobase** | **PPG & ECG** | **Intermediate Fusion (Principal Component Analysis, RMCs)** | **1.39** | **3.7** |
| **RQI-PCA** | **BIDMC** | **PPG & ECG** | **Intermediate Fusion (Principal Component Analysis, RMCs)** | **3.29** | **6.44** |

There are 3 non-fusion methods and 8 fusion methods. The fusion strategy and the fused data of the fusion methods are explained in parentheses below the Estimation Strategy, where the abbreviations are: Electrocardiogram (ECG), Photoplethysmogram (PPG), respiratory rate (RR), intrinsic mode functions (IMFs), respiratory modulated components (RMCs), respiratory quality indices (RQIs).

shows that our method accurately tracked the patient's variable RR, while the other methods struggled with more pronounced fluctuations.

To evaluate the proposed method in other clinical scenarios, we performed an RR prediction comparison using the BIDMC database. Compared to the Capnobase results, the RQI-PCA method performed slightly worse in the BIDMC database, with an ME of -0.09 breaths/min and an MAE of 3.29 breaths/min.

The RQI-PCA method still outperforms other methods in predicting RRs across multiple datasets. Table 2 compares its performance with various methods from the literature, including both non-fusion and fusion approaches. The proposed RQI-PCA method achieves a 17.20% reduction in MAE for RR estimation on the BIDMC dataset and a 24.95% improvement in accuracy on the CapnoBase dataset compared to the existing methods.

As illustrated in Table 2, the RQI-PCA method outperforms several non-fusion methods [8,44,45] in RR estimation accuracy, as demonstrated by results from both the Capnobase and BIDMC datasets. Additionally, comprehensive analyses indicate that methodologies using a single signal generally achieve inferior accuracy compared to multi-modal approaches in RR estimation tasks.

As for methods using fusion strategies, some studies extracted respiratory features from ECG/PPG signals and then applied intermediate or late fusion for RR estimation [20,27,46]. Compared to these methods, the RQI-PCA method excelled. Although the SmartFusion method [20] showed low estimation errors, the window loss it caused remains a notable limitation. Few studies have reported RR estimation algorithms using both ECG and PPG simultaneously [47]. These fusion methods included RRs fusion [48], deep learning [49,50] and respiratory components/futures fusion strategies [34]. Like SmartFusion, Ding *et al.*'s method [48] discarded data with poorer results, leading to a lower error level than RQI-PCA. Although the two deep learning methods [49,50] outperformed the RQI-PCA method, they

require large labeled datasets and substantial computational resources, making them less suitable for wearable respiratory monitoring. In conclusion, the RQI-PCA method offers distinct advantages for continuous RR monitoring.

## 4 Discussion

Abnormal respiratory rates are critical indicators for predicting cardiac arrest and other acute events [51]. In spontaneously breathing patients, RR is often assessed visually by healthcare professionals, a method known to be imprecise [52]. Techniques like spirometry, capnography, or pulmonary compliance assessment can cause patient discomfort, limiting long-term continuous monitoring. Thus, developing methods for continuous, accurate RR measurement that do not interfere with the patient is essential for clinical and daily use. The proposed method uses wearable signals - ECG and PPG - offering a non-invasive solution that reduces patient burden while supporting wearable devices.

Respiration causes changes in thoracic pressure, affecting vasodilation and cardiac output, which can be observed in the PPG signal. However, PPG alone suffers from limited respiratory information content due to susceptibility to motion artifacts, ambient light variations, and sensor displacement, all of which degrade RR estimation accuracy [53]. Similarly, while respiration indirectly affects ECG through RR interval modulation, ECG inherently carries sparse respiratory-related features, with weak signal variations and susceptibility to electrophysiological noise [54]. Consequently, single-signal approaches (PPG- or ECG-only) face fundamental limitations in capturing comprehensive respiratory dynamics, as evidenced by the significantly lower accuracy in Table 2. To address this, we fused ECG and PPG signals in the time domain after RQI screening (ratio ≈1:1). This dual-signal integration compensates for the incomplete respiratory information in individual modalities: PPG provides direct vasomotor responses, while ECG supplements with cardiac rhythm modulations. By synergizing their complementary strengths, the fusion mitigates motion-induced distortions and enhances robustness, as validated by our results.

Some studies show that different window durations have an impact on the model performance. Some studies used window lengths of 32s [34], 60s [10], 64s [24], etc., and most of the window durations were between 30-90s. In this regard, we calculated the MAE obtained for different window lengths (from 30s to 60s) (details can be found in Fig 8). Despite some fluctuations, there was a general trend of increasing MAE as the window length increased. Especially in the BIDMC database, a substantial increase in the error of respiratory rate estimation was observed with the lengthening of the window. This could be attributed to the influence of various factors on respiratory rate, such as physical activity level and emotional state. Longer time windows may encompass a greater variety of changes in breathing patterns, rendering predictions more intricate. While a shorter window length generally yields better estimation results, it's important to note that the shortest window length may not always be optimal. On the one hand, the respiration frequency is 0.05 - 0.75 Hz (3 breaths/min to 45 breaths/min), and we should make sure that there are at least several complete respiration cycles in the window. On the other hand, a very short window length may lead to an increased probability of spectral leakage or frequency folding, which makes the model less robust. In some specific scenarios (e.g., acute respiratory distress), the patient's RR may change dramatically in a short period . Then, it is necessary to use a sliding window of a certain length, which makes it possible to achieve more real-time RR estimation. The sliding step size cannot be too large, because too large a step size is not very meaningful for real-time estimation. Nor can the sliding step size be too small, which would increase the computational and memory burden of the computer. To determine the appropriate window sliding step size, the sliding step size was set as

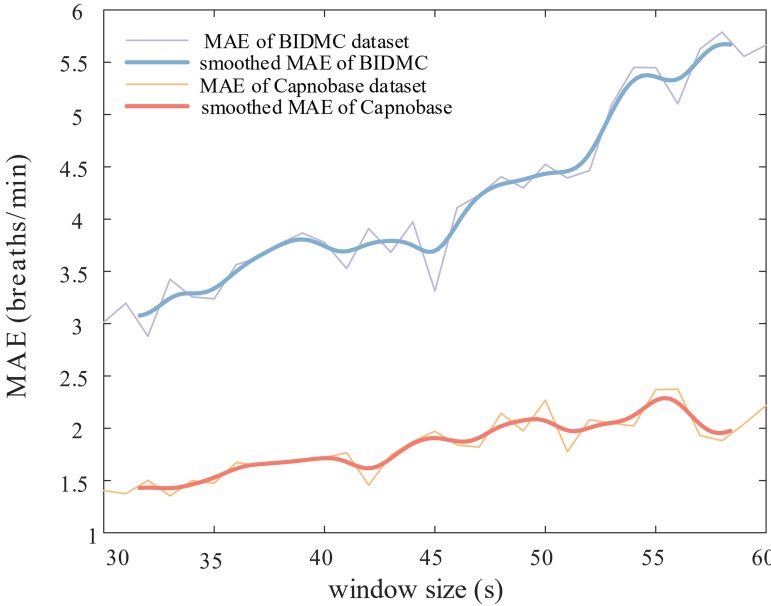

**Fig 8. Variation of error with window length.** The mean absolute error (MAE) of two datasets - BIDMC (purple and blue lines) and Capnobase (yellow and orange lines) - at different window sizes.

the window length (30 s) * n% (n = 10, 20, 30, ..., 100) for the estimation accuracy of respiration rate. The results showed that the estimation error fluctuated slightly with the increase of the window sliding step, but the overall difference was not significant. In the process of practical application, a suitable sliding step should be selected by considering the comprehensive factors such as application scenarios and computational costs.

We tested the accuracy of the proposed temporal fusion method using the Capnobase and BIDMC databases, which contain data from real clinical scenarios: surgical anesthesia (Capnobase), and intensive care (BIDMC). The data encompasses diverse characteristics, including different genders, ages, and conditions. In our study, the MAE accuracy was below 3.5 breaths/min (under 1.5 breaths/min for the Capnobase database), and the method effectively monitored dramatic fluctuations in respiratory rate. This demonstrates the method's adaptability and generalizability for clinical applications.

Our study still lacks data from daily life scenarios. To enhance the generalizability and practicality of our research, we aim to collect or acquire natural respiratory rate data from non-clinical environments. This will provide insights into changes in breathing patterns across different contexts, improving model accuracy and reliability in real-world applications. Future work will focus on incorporating data from daily life, adaptively selecting RMCs, and improving algorithm efficiency for wearable devices.

## 5 Conclusion

In this study, we present a method for continuous respiratory rate (RR) monitoring based on the temporal fusion of various respiratory modulation components extracted from both ECG and PPG signals. Multiple respiratory modulation signals were screened by RQI and then fused by PCA to obtain continuous respiratory signals. Compared with the single signal method and other fusion methods, the proposed method has good accuracy and computational efficiency on BIDMC and Capnobase databases. Future work will focus on expanding

applicability to different populations. In general, our approach enhances existing techniques for non-intrusive RR estimation, offering significant potential for wearable, ubiquitous health monitoring across diverse clinical settings.

## Author contributions

**Conceptualization:** Xiaorong Ding.

**Funding acquisition:** Xiaorong Ding.

**Methodology:** Yan Zhao.

**Validation:** Yuxuan Lin.

**Visualization:** Yuxuan Lin, Xinyue Song.

**Writing – original draft:** Yuxuan Lin, Xinyue Song.

**Writing – review & editing:** Chunlin Zhang, Xiaorong Ding.

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
