## [Decision Letter · Decision Letter 0]

17 Feb 2025

PONE-D-24-54667Continuous Respiratory Rate Monitoring Through Temporal Fusion of ECG and PPG SignalsPLOS ONE

Dear Dr. Ding,

Thank you for submitting your manuscript to PLOS ONE. After careful consideration, we feel that it has merit but does not fully meet PLOS ONE’s publication criteria as it currently stands. Therefore, we invite you to submit a revised version of the manuscript that addresses the points raised during the review process.

We look forward to receiving your revised manuscript.

Kind regards,

Agnese Sbrollini

Academic Editor

PLOS ONE

“This work was supported by National Natural Science Foundation of China (82102178), and in part by Sichuan Science and Technology Program (2021YFH0179).”

Reviewers' comments:

Reviewer's Responses to Questions

**Comments to the Author**

1. Is the manuscript technically sound, and do the data support the conclusions?

Reviewer #1: Partly

Reviewer #2: Yes

2. Has the statistical analysis been performed appropriately and rigorously? 

Reviewer #1: No

Reviewer #2: Yes

3. Have the authors made all data underlying the findings in their manuscript fully available?

Reviewer #1: No

Reviewer #2: Yes

4. Is the manuscript presented in an intelligible fashion and written in standard English?

Reviewer #1: Yes

Reviewer #2: No

5. Review Comments to the Author

Reviewer #1: The authors need to modify the manuscript as per the following :

1. Please specify the new contributions of this work towards the end of Introduction section.

2. Please provide the mathematical expressions of the steps under Methods, with values of parametric settings.

3. Modify the Fig. 2 into a simple signal processing flow diagram with computational blocks connected by arrows. For replication by a reader, please use the notations of variables besides the input output arrow to have a clear understanding.

4. Please cross check equations (2) and (4). Seem to be identical. There is a format error, too.

5. Mention the length of processing window.

6. 1.3.1. Which IMFs were processed? What is the reason for their selection?

7. How PCA was used for fusing Temporal respiratory signal fusion?

8. Table 4: pleas show the results which used the public datasets like BIDMC and CapnoBase. Otherwise, the comparison is 'incompatible'. There are few cited methods using a single physiological signal reporting less error.

9. Conclusion is too brief. Please mention the limitations (if any) of this work and salient outcome.

Reviewer #2: In this work, temporal fusion of respiratory modulated components (RMC) extracted from ECG and PPG signals to estimate respiratory rate (RR) is proposed. The proposed method extracts six RMCs from electrocardiogram (ECG) and photoplethysmogram (PPG). These are fused into one respiratory signal with principal component analysis. Then from the fused signal, the RR is estimated.

The manuscript is interesting; however, the following comments need to be addressed carefully:

1 – In the abstract, there are some typos such as “continous” and “estimaiton”. Check the abstract carefully and correct errors .

2 – In the abstract, include improvement ratio between the proposed and existing works .

3 – In the introduction, include paragraph about feature extraction. There are several feature extraction ne to be included for example, orthogonal polynomials and other tools. See:

[R1] "Orthogonal Polynomials and Their Engineering Applications." International Conference on Business Data Analytics. Cham: Springer Nature Switzerland, 2023.

[R2] "Fast overlapping block processing algorithm for feature extraction." Symmetry 14.4 (2022): 715.

[R3] "High-Performance Krawtchouk Polynomials of High Order Based on Multithreading." Computation 12.6 (2024): 115.

4 – Equation from other sources need to be credited.

5 – Include pseudo code for the proposed method. This will make it more clear to the readers .

6 – include recent work regarding health care such as:

[R4] "Patient monitoring system based on internet of things: A review and related challenges with open research issues." IEEE Access (2024).

[R5] "Internet of Things for in-home health monitoring systems: Current advances, challenges and future directions." IEEE Journal on Selected Areas in Communications 39.2 (2021): 300-310.

7 – There are some typos and grammatical error. Please check and correct them .

- - - - - - - - - - - - - - - - - - - - - - - - - - - - - - - - - - - - - - - - - - - - - - - - - - - - - - - - - - - - - - - - - - - - - - - - - - - - - - - - - - - - - - - - - - - - - - - - - - - - - - - - - - - - - - - - - - - - - - - - - - - - - - - - - - - - - - - - - - - - - - - - - - - - - - - - - - - - - - - - - - - - - - - - - - - - - - - - - - - - - - - - - - - - - - - - - - - - - - - - - - - - - - - - - - - - - - - - - - - - - - - - - - - - - - - - - - - - - - - - - - - - - - - - - - - - - - - - - - - - - - - - - - - - - - - - - - - - - - - - - - - - - - - - - - - - - - - - - - - - - - - - - - - - - - - - - - - - - - - - - - - - - - - - - - - - - - - - - - - - - - - - - - - - - - - - - - - - - - - - - - - - - - - - - - - - - - - - - - - - - - - - - - - - - - - - - - - - - - - - - - - - - - - - - - - - - - - - - - - - - - - - - - - - - - - - - - - - - - - - - - - - - - - - - - - - - - - - - - - - - - - - - - - - - - - - - - - - - - - - - - - - - - - - - - - - - - - - - - - - - - - - - - - - - - - - - - - - - - - - - - - - - - - - - - - - - - - - - - - - - - - - - - - - - - - - - - - - - - - - - - - - - - - - - - - - - - - - - - - - - - - - - - - - - - - - - - - - - - - - - - - - - - - - - - - - - - - - - - - - - - - - - - - - - - - - - - - - - - - - - - - - - - - - - - - - - - - - - - - - - - - - - - - - - - - - - - - - - - - - - - - - - - - - - - - - - - - - - - - - - - - - - - - - - - - - - - - - - - - - - - - - - - - - - - - - - - - - - - - - - - - - - - - - - - - - - - - - - - - - - - - - - - - - - - - - - - - - - - - - - - - - - - - - - - - - - - - - - - - - - - - - - - - - - - - - - - - -

6. PLOS authors have the option to publish the peer review history of their article (what does this mean?). If published, this will include your full peer review and any attached files.

Reviewer #1: No

Reviewer #2: No

---

## [Author Response · Author response to Decision Letter 1]

26 Mar 2025

Dear Editor and Referees,

We sincerely thank you for your constructive comments on our manuscript entitled “Continuous Respiratory Rate Monitoring Through Temporal Fusion of ECG and PPG Signals” (PONE-D-24-54667) with the following co-authors, Yuxuan Lin, Xinyue Song, Yan Zhao, Chunlin Zhang, and Xiaorong Ding. The corresponding author’s email is xiaorong.ding@uestc.edu.cn. We would like to thank all the referees for their careful review of our manuscript and for providing us with valuable comments and suggestions. Your feedback allowed us to substantially improve the quality of the manuscript. Our point-by-point response to the Referees’ comments and a list of the amendments that we made to address their constructive criticisms are shown in BLUE, together with the Referees’ original comments. All the changes are highlighted in RED in the revised manuscript.

Reviewer #1:

The authors need to modify the manuscript as per the following :

Response: The authors would like to express deep appreciation for the referee’s insightful review of our article.

1. Please specify the new contributions of this work towards the end of Introduction section.

Response: We appreciate your suggestion to clarify the novel contributions of this work. As recommended, we have revised the concluding paragraph of the Introduction section (Page 3, Lines 60-68) to explicitly highlight the key advancements of our research, which is quoted for your reference:

“The main contributions of this study are:

We employed both PPG and ECG signals to extract respiration-related components and subsequently fuse them in the time domain. This approach has the potential to yield more comprehensive respiratory-related information compared to the fusion of multiple respiratory rate estimations;

We evaluated the proposed method on two publicly available datasets, Capnobase and BIDMC, with experimental results demonstrating its superior performance over both single-modality approaches and frequency-domain fusion methods.”

2. Please provide the mathematical expressions of the steps under Methods, with values of parametric settings.

Response: Thank you for your constructive feedback. In response to your suggestion, we have incorporated a comprehensive pseudocode in the revised manuscript (Page 4), as illustrated in the accompanying figure, to provide a clear and systematic representation of the algorithm workflow. The pseudocode explicitly outlines the core computational procedures, including the essential mathematical formulations and corresponding parameter configurations employed in our methodology.

3. Modify the Fig. 2 into a simple signal processing flow diagram with computational blocks connected by arrows. For replication by a reader, please use the notations of variables besides the input output arrow to have a clear understanding.

Response: We are grateful for your valuable comments and constructive suggestions. In response to your suggestions, we have thoroughly revised Fig. 2, transforming it into a more streamlined signal processing workflow diagram. The revised figure now features clearly defined computational modules interconnected with directional arrows to explicitly demonstrate the data flow pathways. Furthermore, to enhance the clarity and reproducibility of our methodology, we have incorporated detailed annotations of both input and output variables at each processing stage. The modified figure is demonstrated here for your information:

4. Please cross check equations (2) and (4). Seem to be identical. There is a format error, too.

Response: We sincerely appreciate your thorough review and valuable feedback. Upon careful re-examination, we have identified and corrected several errors in the equation compilation process. The modified equations are shown in Lines 193-196 on Page 7, and also quoted here for your reference:

“Standard Deviation (STD) of Error:

STD=\sqrt{\sum_{i=1}^{n}\frac{\left[\left({RR}_{est}\left(i\right)-{RR}_{ref}\left(i\right)\right)-ME\right]^2}{n-1}},#\left(3\right)

Mean Absolute Error (MAE):

MAE=\frac{\sum_{i=1}^{n}\left|RR_{est}\left(i\right)-RR_{ref}\left(i\right)\right|}{n}#\left(4\right)

In these equations, {RR}_{est} represents the estimated RR, and {RR}_{ref} signifies the reference RR obtained from a validated source.”

Additionally, we have:

• Unified all equation font styles.

• Rechecked all equation cross-references to prevent similar issues.

5. Mention the length of processing window.

Response: Thank you for raising this important point. We have now explicitly specified the processing window duration in the Methods section to improve clarity. Specifically:

Added in Section 1.2 (Page 5, Lines 91-92):

“The raw signal was segmented into 30-second windows to balance accuracy and computational efficiency.”

Additionally, we discussed the importance of processing window length on Page 11 Lines 315 – 320, which is also quoted for your reference:

“Some studies show that different window durations have an impact on the model performance. Some studies used window lengths of 32s [34], 60s [10], 64s [24], etc., and most of the window durations were between 30 - 90s. In this regard, we calculated the MAE obtained for different window lengths (from 30s to 60s) (details can be found in Fig 8). Despite some fluctuations, there was a general trend of increasing MAE as the window length increased. ”

6. 1.3.1. Which IMFs were processed? What is the reason for their selection?

Response: Thank you for your careful review. Regarding your inquiry, we have supplemented the following figure and explanations in Section 1.3.1 (Pages 5-6 Lines 126-140) of the manuscript, which we quote here:

Rationale for IMF Selection:

“RMCs have a narrow frequency band and fewer harmonics, which belong to the ‘spectrum sparse’ signal, and a small number of IMFs can be selected. Through systematic parametric optimization experiments, we determined that the VMD level of 5 optimally balances the extraction of the signal’s primary time-frequency components while avoiding mode mixing (see Fig 3). This configuration ensures the preservation of the original signal’s physical interpretability. ”

Key IMF Analysis:

“Subsequently, we calculated the maximum peak frequency within the power spectrum of each IMF. The IMF with the smallest absolute deviation from the peak frequency of the input signal’s power spectrum was selected as the representative filtered respiratory component. Fig 3 shows an example of VMD filtering, the spectral peak of IMF3 (0.3 Hz) exhibits exact correspondence with the characteristic frequency of the original signal. In contrast, IMF1 and IMF2 predominantly consist of high-frequency noise components, whereas IMF4 and IMF5 represent low-frequency residual artifacts. Based on this spectral alignment and component characterization, IMF3 was rigorously identified as the critical modality for subsequent analysis, ensuring fidelity to the signal’s intrinsic features.”

7. How PCA was used for fusing Temporal respiratory signal fusion?

Response: We sincerely appreciate the reviewer's insightful comment regarding the application of PCA for temporal respiratory signal fusion. Below, we provide a detailed clarification of our methodology:

PCA was employed as an optimal linear dimensionality reduction technique to fuse respiratory motion signals (RMCs) while preserving maximum variance. The selection of PCA is motivated by its ability to 1) eliminate redundant information across multiple respiratory signals, 2) automatically weight dominant respiratory patterns through eigenvalue decomposition, and 3) generate a unified respiratory waveform (RW_{est}) with minimal information loss.

Additionally, our implementation rigorously follows these steps (on Page 6 Lines 172-180, as quoted below):

“PCA can reduce the M-dimensional information in X space to K-dimensional information in Y space (K<M) with less information loss. Assuming that there are m RMCs after RQI screening, the steps of PCA fusion are as follows:

(1) Form matrix \mathbit{X} by arranging the n RMCs as columns.

Center the matrix \mathbit{X} (mean subtraction).

Compute the covariance matrix \mathbit{P} and calculate its eigenvalues and eigenvectors.

Sort the eigenvectors into a matrix in descending order of their corresponding eigenvalues, then take the first row to form matrix \mathbit{Q}.

Compute \mathbit{Y}=\mathbit{QX}, which yields the fused respiratory waveform RW_{est}.”

The first principal component inherently weights respiratory phases (inspiration/expiration) proportionally to their variance contributions, effectively suppressing outlier artifacts while enhancing consistent respiratory patterns.

8. Table 4: pleas show the results which used the public datasets like BIDMC and CapnoBase. Otherwise, the comparison is 'incompatible'. There are few cited methods using a single physiological signal reporting less error.

Response: We sincerely appreciate your valuable feedback and constructive suggestions. In accordance with your suggestion regarding databases, we have revised Table 2 to exclusively include results from widely recognized public datasets (BIDMC and CapnoBase) to enhance reproducibility and comparability across studies (supplemented the following figure). Studies utilizing non-public or proprietary datasets have been removed. The tabular organization was optimized through strategic rearrangement guided by the Estimation Strategy, improving visual coherence with our methodological architecture.

Furthermore, in response to your feedback regarding methods utilizing a single signal, we have enhanced the Discussion section (Page 11 Lines 290-310) to clarify why these methods tend to yield higher errors, which we include here for your reference:

“However, PPG alone suffers from limited respiratory information content due to susceptibility to motion artifacts, ambient light variations, and sensor displacement, all of which degrade RR estimation accuracy [53]. Similarly, while respiration indirectly affects ECG through RR interval modulation, ECG inherently carries sparse respiratory-related features, with weak signal variations and susceptibility to electrophysiological noise [54]. Consequently, single-signal approaches (PPG- or ECG-only) face fundamental limitations in capturing comprehensive respiratory dynamics, as evidenced by the significantly lower accuracy in Table 2. To address this, we fused ECG and PPG signals in the time domain after RQI screening (ratio 1:1). This dual-signal integration compensates for the incomplete respiratory information in individual modalities: PPG provides direct vasomotor responses, while ECG supplements with cardiac rhythm modulations. By synergizing their complementary strengths, the fusion mitigates motion-induced distortions and enhances robustness, as validated by our results.”

9. Conclusion is too brief. Please mention the limitations (if any) of this work and salient outcome.

Response: Thank you for your valuable feedback on our manuscript. We have carefully considered your comments and expanded the conclusion (Page 12 Lines 359-368) to include a more detailed discussion of the limitations of our work and the salient outcomes as quoted below:

“In this study, we present a method for continuous RR monitoring based on the temporal fusion of various respiratory modulation components extracted from both ECG and PPG signals. Multiple respiratory modulation signals were screened by RQI and then fused by PCA to obtain continuous respiratory signals. Compared with the single signal method and other fusion methods, the proposed method has good accuracy and computational efficiency on BIDMC and Capnobase databases. Future work will focus on expanding applicability to different populations. In general, our approach enhances existing techniques for non-intrusive RR estimation, offering significant potential for wearable, ubiquitous health monitoring across diverse clinical settings.”

We hope these revisions address your concerns. Please let us know if there are any further adjustments needed.

Reviewer #2:

In this work, temporal fusion of respiratory modulated components (RMC) extracted from ECG and PPG signals to estimate respiratory rate (RR) is proposed. The proposed method extracts six RMCs from electrocardiogram (ECG) and photoplethysmogram (PPG). These are fused into one respiratory signal with principal component analysis. Then from the fused signal, the RR is estimated.

The manuscript is interesting; however, the following comments need to be addressed carefully:

Response: Thank you sincerely for your positive assessment of our work's significance and your constructive feedback. We have carefully addressed each of your comments as follows.

1 – In the abstract, there are some typos such as “continous” and “estimaiton”. Check the abstract carefully and correct errors .

Response: Thank you for your meticulous review. We have thoroughly checked and corrected the typographical errors in the abstract:

"continous" revised to "continuous" (Page 1, Abstract)

"estimaiton" revised to "estimation" (Page 1, Abstract)

To ensure linguistic accuracy, we have implemented:

Full-text grammar check via Grammarly;

Manual sentence-by-sentence proofreading;

Terminology verification against PLOS's Guidelines for Scientific Writing.

All linguistic revisions in the abstract are highlighted in red in the revised manuscript.

2 – In the abstract, include improvement ratio between the proposed and existing works .

Response: We sincerely appreciate the reviewer's insightful suggestion. Following this valuable comment, we have revised the abstract to explicitly state the performance improvement ratios. Specifically, in the last paragraph of the abstract (Page 1, Abstract), we now quote for your reference:

“achieving an 11.61% reduction in MAE compared to existing state-of-the-art approaches.”

Additionally, we added a more detailed comparative analysis in the Results section (Page 9 Lines 257-260), which is quoted here:

“The proposed RQI-PCA method achieves a 17.20% reduction in MAE for RR estimation on the BIDMC dataset and a 24.95% improvement in accuracy on the CapnoBase dataset compared to the existing methods.”

3 – In the introduction, include paragraph about feature extraction. There are several feature extraction ne to be included for example, orthogonal polynomials and other tools. See:

[R1] "Orthogonal Polynomials and Their Engineering Applications." International Conference on Business Data Analytics. Cham: Springer Nature Switzerland, 2023.

[R2] "Fast overlapping block processing algorithm for feature extraction." Symmetry 14.4 (2022): 715.

[R3] "High-Performance Krawtchouk Polynomials of High Order Based on Multithreading." Computation 12.6 (2024): 115.

Response: Thank you for highlighting the importance of properly contextualizing feature extraction methodologies. We have carefully integrated references to the seminal works on orthogonal polynomials [R1,R3] and block processing algorithms [R2] throughout the Introduction section (references 13-15) and added them on Page 2 Lines 19-20, which is quoted below:

“Recent advancements in signal processing algorithms have significantly enhanced computational efficiency [13][14][15]).”

4 – Equation from other sources need to be credited.

Response: We appreciate your attention to the proper attribution of equations derived from other sources. In response to your suggestion, we have carefully reviewed all equations in our manuscript and made the following revisions.

For equations that were adapted or directly cited from other studies, we have now explicitly acknowledged the sources in the text (Page 6 Lines 149-154), which is quoted below for your reference:

“The standard RQI calculation uses Fourier transform to define the energy concentration of a specific frequency band of the RMC [34]:

RQI= PmaxPall,#1

where P_{max} is the sum of the spectral coefficients within the physiological respiration range of 0.05 to 0.75 Hz (equivalent to 3 - 45 breaths/min), and P_{all} is the sum of Fourier transform coefficients across the entire spectrum.”

5 – Include pseudo code for the proposed method. This will make it more clear to the readers.

---

## [Decision Letter · Decision Letter 1]

12 May 2025

Continuous Respiratory Rate Monitoring Through Temporal Fusion of ECG and PPG Signals

PONE-D-24-54667R1

Dear Dr. Ding,

We’re pleased to inform you that your manuscript has been judged scientifically suitable for publication and will be formally accepted for publication once it meets all outstanding technical requirements.

Kind regards,

Agnese Sbrollini

Academic Editor

PLOS ONE

Additional Editor Comments (optional):

Reviewers' comments:

Reviewer's Responses to Questions

**Comments to the Author**

1. If the authors have adequately addressed your comments raised in a previous round of review and you feel that this manuscript is now acceptable for publication, you may indicate that here to bypass the “Comments to the Author” section, enter your conflict of interest statement in the “Confidential to Editor” section, and submit your "Accept" recommendation.

Reviewer #2: All comments have been addressed

Reviewer #3: All comments have been addressed

2. Is the manuscript technically sound, and do the data support the conclusions?

Reviewer #2: Yes

Reviewer #3: Yes

3. Has the statistical analysis been performed appropriately and rigorously? 

Reviewer #2: Yes

Reviewer #3: Yes

4. Have the authors made all data underlying the findings in their manuscript fully available?

Reviewer #2: Yes

Reviewer #3: Yes

5. Is the manuscript presented in an intelligible fashion and written in standard English?

Reviewer #2: Yes

Reviewer #3: Yes

6. Review Comments to the Author

Reviewer #2: In this work, temporal fusion of respiratory modulated components (RMC) extracted from ECG and PPG signals to estimate respiratory rate (RR) is proposed. The proposed method extracts six RMCs from electrocardiogram (ECG) and photoplethysmogram (PPG). These are fused into one respiratory signal with principal component analysis. Then from the fused signal, the RR is estimated.

The authors have addressed the raised comments.

- - - - - - - - - - - - - - - - - - - - - - - - - - - - - - - - - - - - - - - - - - - - - - - - - - - - - - - - - - - - - - - - - - - - - - - - - - - - - - - - - - - - - - - - - - - - - - - - - - - - - - - - - - - - - - - - - - - - - - - - - - - - - - - - - - - - - - - - - - - - - - - - - - - - - - - - - - - - - - - - - - - - - - - - - - - - - - - - - - - - - - - - - - - - - - - - - - - - - - - - - - - - - - - - - - - - - - - - - - - - - - - - - - - - - - - - - - - - - - - - - - - - - - - - - - - - - - - - - - - - - - - - - - - - - - - - - - - - - - - - - - - - - - - - - - - - - - - - - - - - - - - - - - - - - - - - - - - - - - - - - - - - - - - - - - - - - - - - - - - - - - - - - - - - - - - - - - - - - - - - - - - - - - - - - - - - - - - - - - - - - - - - - - - - - - - - - - - - - - - - - - - - - - - - - - - - - - - - - - - - - - - - - - - - - - - - - - - - - - - - - - - - - - - - - - - - - - - - - - - - - - - - - - - - - - - - - - - - - - - - - - - - - - - - - - - - - - - - - - - - - - - - - - - - - - - - - - - - - - - - - - - - - - - - - - - - - - - - - - - - - - - - - - - - - - - - - - - - - - - - - - - - - - - - - - - - - - - - - - - - - - - - - - - - - - - - - - - - - - - - - - - - - - - - - - - - - - - - - - - - - - - - - - - - - - - - - - - - - - - - - - - - - - - - - - - - - - - - - - - - - - - - - - - - - - - - - - - - - - - - - - - - - - - - - - - - - - - - - - - - - - - - - - - - - - - - - - - - - - - - - - - - - - - - - - - - - - - - - - - - - - - - - - - - - - - - - - - - - - - - - - - - - - - - - - - - - - - - - - - - - - - - - - - - - - - - - - - - - - - - - - - - - - - - - - - - - - - -

Reviewer #3: Reading the paper and the comments of previous reviewers it appears all previous comments have been addressed

7. PLOS authors have the option to publish the peer review history of their article (what does this mean?). If published, this will include your full peer review and any attached files.

Reviewer #2: No

Reviewer #3: No

---

## [Editor Report · Acceptance letter]

PONE-D-24-54667R1

PLOS ONE

Dear Dr. Ding,

I'm pleased to inform you that your manuscript has been deemed suitable for publication in PLOS ONE. Congratulations! Your manuscript is now being handed over to our production team.

Kind regards,

on behalf of

Dr. Agnese Sbrollini

Academic Editor

PLOS ONE